# Catching heuristics are optimal control policies

**Boris Belousov**[*], **Gerhard Neumann**[*], **Constantin A. Rothkopf**[**], **Jan Peters**[*]

[*]Department of Computer Science, TU Darmstadt
[**]Cognitive Science Center & Department of Psychology, TU Darmstadt

## Abstract

Two seemingly contradictory theories attempt to explain how humans move to intercept an airborne ball. One theory posits that humans *predict* the ball trajectory to *optimally* plan future actions; the other claims that, instead of performing such complicated computations, humans employ *heuristics* to reactively choose appropriate actions based on *immediate* visual feedback. In this paper, we show that interception strategies appearing to be heuristics can be understood as computational solutions to the optimal control problem faced by a ball-catching agent acting under uncertainty. Modeling catching as a continuous partially observable Markov decision process and employing stochastic optimal control theory, we discover that the four main heuristics described in the literature are optimal solutions if the catcher has sufficient time to continuously visually track the ball. Specifically, by varying model parameters such as noise, time to ground contact, and perceptual latency, we show that different strategies arise under different circumstances. The catcher's policy switches between generating reactive and predictive behavior based on the ratio of system to observation noise and the ratio between reaction time and task duration. Thus, we provide a rational account of human ball-catching behavior and a unifying explanation for seemingly contradictory theories of target interception on the basis of stochastic optimal control.

## 1 Introduction

Humans exhibit impressive abilities of intercepting moving targets as exemplified in sports such as baseball [6]. Despite the ubiquity of this visuomotor capability, explaining how humans manage to catch flying objects is a long-standing problem in cognitive science and human motor control. What makes this problem computationally difficult for humans are the involved perceptual uncertainties, high sensory noise, and long action delays compared to artificial control systems and robots. Thus, understanding action generation in human ball interception from a computational point of view may yield important insights on human visuomotor control. Surprisingly, there is no generally accepted model that explains empirical observations of human interception of airborne balls. McIntyre *et al.* [15] and Hayhoe *et al.* [13] claim that humans employ an internal model of the physical world to predict where the ball will hit the ground and how to catch it. Such internal models allow for planning and potentially optimal action generation, e.g., they enable optimal catching strategies where humans predict the interception point and move there as fast as mechanically possible to await the ball. Clearly, there exist situations where latencies of the catching task require such strategies (e.g., when a catcher moves the arm to receive the pitcher's ball). By contrast, Gigerenzer & Brighton [11] argue that the world is far too complex for sufficiently precise modeling (e.g., a catcher or an outfielder in baseball would have to take air resistance, wind, and spin of the ball into account to predict its trajectory). Thus, humans supposedly extract few simple but robust features that suffice for successful execution of tasks such as catching. Here, immediate feedback is employed to guide action generation instead of detailed modeling. Policies based on these features are called *heuristics* and the claim is that humans possess a bag of such tricks, the "adaptive toolbox". For a baseball outfielder, a successful heuristic could be "Fix your gaze on the ball, start running, and adjust your running speed so that the angle of gaze remains constant" [10]. Thus, at the core, finding a unifying computational account of the human interception of moving targets also contributes to the long-lasting debate about the nature of human rationality [20].

In this paper, we propose that these seemingly contradictory views can be unified using a single computational model based on a continuous partially observable Markov decison process model (POMDP). In this model, the intercepting agent is assumed to choose optimal actions that take uncertainty about future movement into account. This model prescribes that both the catcher and the outfielder act optimally for their respective situation and uncertainty. We show that an outfielder agent using a highly stochastic internal model for prediction will indeed resort to purely reactive polices resembling established heuristics from the literature. The intuitive reason for such short-sighted behavior being optimal is that ball predictions over sufficiently long time horizons with highly stochastic models effectively become guessing. Similarly, our model will yield optimally planned actions based on predictions if the uncertainty encountered by the catcher agent is low while the latency is non-negligible in comparison to the movement duration. Moreover, we identify catching scenarios where the only strategy to intercept the ball requires to turn away from it and run as fast as possible. While such strategies cannot be explained by the heuristics proposed so far, the optimal control approach yields a plausible policy exhibiting both reactive and feedforward behavior. While other motor tasks (e.g., reaching movements [9, 22], locomotion [1]) have been explained in terms of stochastic optimal control theory, to the best of our knowledge this paper is the first to explain ball catching within this computational framework. We show that the four previously described empirical heuristics are actually optimal control policies. Moreover, our approach allows predictions for settings that cannot be explained by heuristics and have not been studied before. As catching behavior has previously been described as a prime example of humans not following complex computations but using simple heuristics, this study opens an important perspective on the fundamental question of human rationality.

## 2   Related work

A number of heuristics have been proposed to explain how humans catch balls, see [27, 8, 16] for an overview. We focus on three theories well-supported by experiments: Chapman's theory, the generalized optic acceleration cancellation (GOAC) theory, and the linear optical trajectory (LOT) theory.

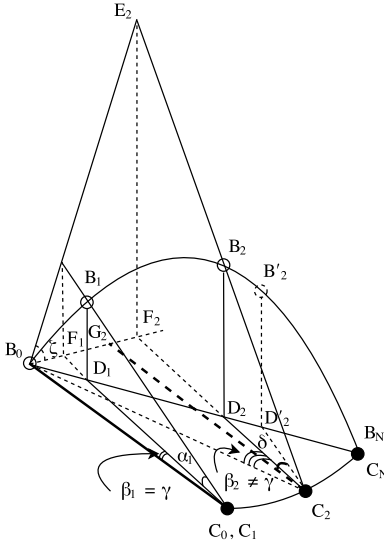

Figure 1: Well-known heuristics.

Chapman [6] considered a simple kinematic problem (see Figure 1) where the ball $B$ follows a parabolic trajectory $B_{0:N}$ while the agent $C$ follows $C_{0:N}$ to intercept it. Only the position of the agent is relevant—his gaze is always directed towards the ball. Angle $\alpha$ is the *elevation angle*; angle $\gamma$ is the *bearing angle* with respect to direction $C_0B_0$ (or $C_2G_2$ which is parallel). Due to delayed reaction, the agent starts running when the ball is already in the air. Chapman proposed two heuristics, i.e., the optic acceleration cancellation (OAC) that prescribes maintaining $\mathrm{d}\tan\alpha/\mathrm{d}t = \text{const}$, and the constant bearing angle (CBA), which requires $\gamma = \text{const}$. However, Chapman did not explain how these heuristics cope with disturbances and observations.

To incorporate visual observations, McLeod *et al.* [16] introduced the *field of view* of the agent into Chapman's theory and coupled the agent's running velocity to the location of the ball in the visual field. Instead of the CBA heuristic, a *tracking heuristic* is employed to form the generalized optic acceleration cancellation (GOAC) theory. This tracking heuristic allows reactions to uncertain observations. In our example in Figure 1, the agent might have moved from $C_0$ to $C_2$ while maintaining a constant $\gamma$. To keep fulfilling this heuristic, the ball needs to arrive at $B_2$ at the same time. However, if the ball is already at $B'_2$, the agent will see it falling into the right side of his field of view and he will speed up. Thus, the agent internally tracks the angle $\delta$ between $CD$ and $C_0B_0$ and attempts to adjust $\delta$ to $\gamma$.

In Chapman's theory and the GOAC theory, the elevation angle $\alpha$ and the bearing angle $\gamma$ are controlled independently. As such, separate control strategies are implausible, therefore McBeath *et al.* [14] proposed the linear optical trajectory (LOT) heuristic that controls both angles jointly. LOT suggests that the catching agent runs such that the projection of the ball trajectory onto the plane perpendicular to the direction $CD$ remains linear, which implies that $\zeta = \angle E_2B_0F_2$ remains constant. As $\tan\zeta = \tan\alpha_2/\tan\beta_2$ can be observed from the pyramid $B_0F_2C_2E_2$ with the right angles at $F_2$, there exist a coupling between the elevation angle $\alpha$ and the horizontal optical angle $\beta$ (defined as the angle between $CB_0$ and $CD$), which can be used for directing the agent.

In contrast to the literature on outfielder's catching in baseball, other strands of research in human motor control have focused on predictive models [17] and optimality of behavior [9, 22]. Tasks similar to the catcher's in baseball have yielded evidence for prediction. Humans were shown to anticipate where a tennis ball will hit the floor when thrown with a bounce [13], and humans also appear to use an internal model of gravity to estimate time-to-contact when catching balls [15]. Optimal control theory has been used to explain reaching movements (with cost functions such as minimum-jerk [9], minimum-torque-change [23] and minimum end-point variance [12]), motor coordination [22], and locomotion (as minimizing metabolic energy [1])

## 3 Modeling ball catching under uncertainty as an optimal control problem

To parsimoniously model the catching agent, we rely on an optimal control formulation (Sec. 3.1) where the agent is described in terms of state-transitions, observations and a cost function (Sec. 3.2).

### 3.1 Optimal control under uncertainty

In optimal control, the interaction of the agent with the environment is described by a stochastic dynamic model or system (e.g., describing ball flight and odometry). The system's state

$$\boldsymbol{x}_{k+1} = \boldsymbol{f}(\boldsymbol{x}_k, \boldsymbol{u}_k) + \boldsymbol{\epsilon}_{k+1}, \quad k = 0 \ldots N - 1, \tag{1}$$

at the next time step $k + 1$ is given as a noisy function of the state $\boldsymbol{x}_k \in \mathbb{R}^n$ and the action $\boldsymbol{u}_k \in \mathbb{R}^m$ at the current time step $k$. The mean state dynamics $\boldsymbol{f}$ are perturbed by zero-mean stationary white Gaussian noise $\boldsymbol{\epsilon}_k \sim \mathcal{N}(\boldsymbol{0}, \boldsymbol{Q})$ with a constant system noise covariance matrix $\boldsymbol{Q}$ modeling the uncertainty in the system (e.g., the uncertainty in the agent's and ball's positions).

The state of the system is not always fully observed (e.g., the catching agent can only observe a ball when he looks at it), lower-dimensional than the system's state (e.g., only ball positions can directly be observed) and the observations are generally noisy (e.g., visuomotor noise affects ball position estimates). Thus, at every time step $k$, sensory input provides a noisy lower-dimensional measurement $\boldsymbol{z}_k \in \mathbb{R}^p$ of the true underlying system state $\boldsymbol{x}_k \in \mathbb{R}^n$ with $p < n$ described by

$$\boldsymbol{z}_k = \boldsymbol{h}(\boldsymbol{x}_k) + \boldsymbol{\delta}_k, \quad k = 1 \ldots N, \tag{2}$$

where $\boldsymbol{h}$ is a deterministic observation function and $\boldsymbol{\delta}_k \sim \mathcal{N}(\boldsymbol{0}, \boldsymbol{R}_k)$ is zero-mean non-stationary white Gaussian noise with a state-dependent covariance matrix $\boldsymbol{R}_k = \boldsymbol{R}(\boldsymbol{x}_k)$. For catching, such state-dependency is crucial to modeling the effect of the human visual field. When the ball is at its center, measurements are least uncertain; whereas when the ball is outside the visual field, observations are maximally uncertain.

The agent obviously can only generate actions based on the observations collected so far, while affecting his and the environment's true next state. The history of observations allows forming probability distributions over the state at different time-steps called *beliefs*. Taking the uncertainty in (1) and (2) into account, the agent needs to plan and control in the *belief space* (i.e., the space of probability distributions over states) rather than in the state space. We approximate belief $\boldsymbol{b}_k$ about the state of the system at time $k$ by a Gaussian distribution with mean $\boldsymbol{\mu}_k$ and variance $\boldsymbol{\Sigma}_k$. For brevity, we write $\boldsymbol{b}_k = (\boldsymbol{\mu}_k, \boldsymbol{\Sigma}_k)$, associating the belief with its sufficient statistics. Belief dynamics $(\boldsymbol{b}_{k-1}, \boldsymbol{u}_{k-1}, \boldsymbol{z}_k) \rightarrow \boldsymbol{b}_k$ is approximated by the extended Kalman filter [21, Chapter 3.3].

A cost function $J$ can be a parsimonious description of the agent's objective. The agent will choose the next action by optimizing such a cost function with respect to all future actions at every time-step. To make the resulting optimal control computations numerically tractable, future observations need to be assumed to coincide with their most likely values (see e.g., [19, 5]). Thus, at every time step, the agent solves a constrained nonlinear optimization problem

$$\begin{aligned} \min_{\boldsymbol{u}_{0:N-1}} \quad & J(\boldsymbol{\mu}_{0:N}, \boldsymbol{\Sigma}_{0:N}; \boldsymbol{u}_{0:N-1}) \\ \text{s.t.} \quad & \boldsymbol{u}_k \in \mathcal{U}_{\text{feasible}}, \quad k = 0 \ldots N - 1, \\ & \boldsymbol{\mu}_k \in \mathcal{X}_{\text{feasible}}, \quad k = 0 \ldots N, \end{aligned} \tag{3}$$

which returns an optimal sequence of controls $\boldsymbol{u}_{0:N-1}$ minimizing the objective function $J$. The agent executes the first action, obtains a new observation, and replans again; such an approach is known as *model predictive control*. The policy resulting from such computations is sub-optimal because of open-loop planning and limited time horizon, but with growing time horizon it approaches the optimal policy. Reaction time $\tau_r$ can be incorporated by delaying the observations. An interesting property of this model is that the catching agent decides on his own in an optimal way when to gather information by looking at the ball and when to exploit already acquired knowledge depending on the level of uncertainty he agrees to tolerate.

## 3.2 A computational model of the catching agent for belief-space optimal control

Here we explain the modeling assumptions concerning states, actions, state transitions, and observations. After that we describe the cost function that the agent has to minimize.

**States and actions.** The state of the system $\boldsymbol{x}$ consists of the location and velocity of the ball in 3D space, the location and velocity of the catching agent in the ground plane, and the agent's gaze direction represented by a unit 3D vector. The agent's actions $\boldsymbol{u}$ consist of the force applied to the center of mass and the rate of change of the gaze direction.

**State transitions and observations.** Several model components are essential to faithfully describe catching behavior. First, the state transfer is described by the damped dynamics of the agent's center of mass $\ddot{\boldsymbol{r}}_c = \boldsymbol{F} - \lambda \dot{\boldsymbol{r}}_c$, where $\boldsymbol{r}_c = [x, y]$ are the agent's Cartesian coordinates, $\boldsymbol{F}$ is the applied force resulting from the agent's actions, and $\lambda$ is the damping coefficient. Damping ensures that the catching agent's *velocity does not grow without bound* when the maximum force is applied. The magnitude of the maximal force and the friction coefficient are chosen to fit Usain Bolt's sprint data[1]. Second, the gaze vector's direction $\boldsymbol{d}$ is controlled through the first derivatives of the two angles that define it. These are the angle between $\boldsymbol{d}$ and its projection onto the $xy$-plane and the angle between $\boldsymbol{d}$'s projection onto the $xy$-plane and the $x$-axis. Such parametrization of the actions allows for realistically *fast changes of gaze direction*. Third, the maximal running speed depends on the gaze direction, e.g., *running backwards is slower* than running forward or even sideways. This relationship can be incorporated through dependence of the maximal applicable force $\boldsymbol{F}_{\max}$ on the direction $\boldsymbol{d}$. It can be expressed by limiting the magnitude of the maximal applicable force $|\boldsymbol{F}_{\max}(\theta)| = F_1 + F_2 \cos\theta$, where $\theta$ is the angle between $\boldsymbol{F}$ (i.e., the direction into which the catcher accelerates) and the projection of the catcher's gaze direction $\boldsymbol{d}$ onto the $xy$-plane. The parameters $F_1$ and $F_2$ are chosen to fit human data on forward and backwards running[2]. The resulting continuous time dynamics of agent and ball are converted into discrete time state transfers using the classical Runga-Kutta method. Fourth, the *observation uncertainty depends on the state*, which reflects the fact that humans' visual resolution falls off across the visual field with increasing distance from the fovea. When the ball falls to the side of the agent's field of view, the uncertainty about ball's position grows according to $\sigma_o^2 = s(\sigma_{\max}^2(1 - \cos\Omega) + \sigma_{\min}^2)$ depending on the distance to the ball $s$ and the angle $\Omega$ between gaze direction $\boldsymbol{d}$ and the vector pointing from the agent towards the ball. The parameters $\{\sigma_{\min}, \sigma_{\max}\}$ control the scale of the noise. The ball is modeled as a parabolic flight perturbed by Gaussian noise with variance $\sigma_b^2$.

**Cost function.** The catching agent has to trade-off success (i.e., catching the ball) with effort. In other words, he aims at *maximizing the probability of catching the ball with minimal effort*. A ball is assumed to be caught if it is within reach, i.e., not further away from the catching agent than $\varepsilon_{\text{threshold}}$ at the final time. Thus, the probability of catching the ball can be expressed as $\Pr(|\boldsymbol{\mu}_b - \boldsymbol{\mu}_c| \leq \varepsilon_{\text{threshold}})$, where $\boldsymbol{\mu}_b$ and $\boldsymbol{\mu}_c$ are the predicted positions of the ball and the agent at the final time (i.e., parts of the belief state of the agent). Since such beliefs are modeled as Gaussians, this probability has a unique global maximum at $\boldsymbol{\mu}_b = \boldsymbol{\mu}_c$ and $\boldsymbol{\Sigma}_N \to \boldsymbol{0}^+$. Therefore, a final cost $J_{\text{final}} = w_0 \|\boldsymbol{\mu}_b - \boldsymbol{\mu}_c\|_2^2 + w_1 \operatorname{tr}\boldsymbol{\Sigma}_N$ can approximate the negated log-probability of successfully catching the ball while rendering the optimal control problem solvable. The weights $w_0$ and $w_1$ are set to optimally approximate this negated log-probability. The desire of the agent to be energy efficient is encoded as a penalty on the control signals $J_{\text{energy}} = \tau \sum_{k=0}^{N-1} \boldsymbol{u}_k^T \boldsymbol{M} \boldsymbol{u}_k$ with the fixed duration $\tau$ of the discretized time steps and a diagonal weight matrix $\boldsymbol{M}$ to trade-off controls. Finally, we add a term that penalizes agent's uncertainty at every time step $J_{\text{running}} = \tau w_2 \sum_{k=0}^{N-1} \operatorname{tr}\boldsymbol{\Sigma}_k$ that encodes the agent preference of certainty over uncertainty. It appears naturally in optimal control problems when the maximum likelihood observations assumption is relaxed [24] and captures how final uncertainty distributes over the preceding time steps, but has to be added explicitly within the model predictive control framework in order to account for replanning at every time step. The complete cost function is thus given by the sum

$$J = J_{\text{final}} + J_{\text{running}} + J_{\text{energy}} = \underbrace{w_0 \|\boldsymbol{\mu}_b - \boldsymbol{\mu}_c\|_2^2}_{\text{final position}} + \underbrace{w_1 \operatorname{tr}\boldsymbol{\Sigma}_N}_{\text{final uncertainity}} + \underbrace{\tau w_2 \sum_{k=0}^{N-1} \operatorname{tr}\boldsymbol{\Sigma}_k}_{\text{running uncertainty}} + \underbrace{\tau \sum_{k=0}^{N-1} \boldsymbol{u}_k^T \boldsymbol{M} \boldsymbol{u}_k}_{\text{total energy}}$$

that the catching agent has to minimize in order to successfully intercept the ball.

### 3.3 Implementation details

To solve Problem (3), we use the covariance-free multiple shooting method [18] for trajectory optimization [7, 3] in the belief space. Derivatives of the cost function are computed using CasADi [2]. Non-linear optimization is carried out by Ipopt [26]. L-BFGS and warm-starts used.

## 4 Simulated experiments and results

In this section, we present the results of two simulated scenarios and a comparative evaluation. First, using the optimal control approach, we show that continuous tracking (where the ball always remains in the field of view of the outfielder) naturally leads to the heuristics from literature [6, 16, 14] if the catching agent is sufficiently fast in comparison to the ball independent of whether he is running forward, backwards, or sideways. Subsequently, we show that more complex behavior arises when the ball is too fast to be caught while running only sideways or backwards (e.g., as in soccer or long passes in American football). Here, tracking is interrupted as the agent needs to turn away from the ball to run forward. While the heuristics break, our optimal control formulation exhibits plausible strategies similar to those employed by human catchers. Finally, we systematically study the effects of noise and time delay onto the agent's policy. The optimal control policies arising from our model switch between reactive and predictive behaviors depending on uncertainty and latency.

### 4.1 Continuous tracking of an outfielder—heuristics hold

To directly compare our model against empirical catching data that has been described as resulting from a heuristic, we reproduce the settings from [16] where a ball flew $15\,\mathrm{m}$ in $3\,\mathrm{s}$ and a human subject starting about $6\,\mathrm{m}$ away from the impact point had to intercept it.

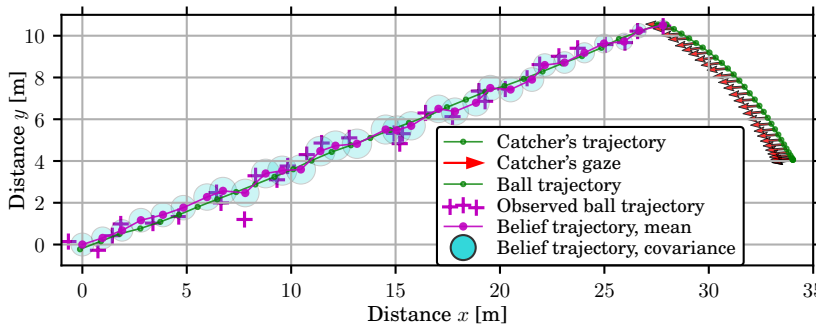

The optimal control policy can deal with such situations and yields the behavior observed by McLeod *et al.* [16]. In fact, even when doubling all distances the reactive control policy exhibits all four major heuristics (OAC, GOAC, CBA and LOT) with approximately the same precision as in the original human experiments. Figure 2 shows a typical simulated catch viewed from above. The ball and the agent's true trajectories are depicted in green (note that the ball is frequently hidden behind the belief state trajectory). The agent's observations and the mean belief trajectory of the ball are represented by magenta crosses and a magenta line, respectively. The belief uncertainty is indicated by the cyan ellipsoids that capture 95% of the probability mass. The gaze vectors of the agent are shown as red arrows. The catching agent starts sufficiently close to the interception point to continuously visually track the ball, therefore he is able to efficiently reduce his uncertainty on the ball's position and successfully intercept it while keeping it in sight. Note that the agent does not follow a straight trajectory but a curved one in agreement with human experiments [16].

Figure 2: A typical simulated trajectory of a successful catch in the continuous tracking scenario as encountered by the outfielder. The uncertainty in the belief state is kept low by the agent by fixating the ball. Such empirically observed scenarios [6, 16, 14] have led to the proposition of the heuristics which arise naturally from our optimal control formulation.

Figure 3 shows plots of the relevant angles over time to compare the behavior exhibited by human catchers to the optimal catching policy. The tangent of the elevation angle $\tan\alpha$ grows linearly with time, as predicted by the optic acceleration cancellation heuristic (OAC). The bearing angle $\gamma$ remains constant (within a $5\,\mathrm{deg}$ margin) as predicted by the constant bearing angle heuristic (CBA). The rotation angle $\delta$ oscillates around $\gamma$ as predicted by the generalized optic acceleration cancellation theory (GOAC). The tangent of the horizontal optical angle $\tan\beta$ is proportional to $\tan\alpha$, as predicted by the linear optical trajectory theory (LOT). The small oscillations in the rotation angle and in the horizontal optical angle are due to reaction delay and uncertainty; they are also predicted by GOAC and LOT. Thus, in this well-studied case, the model produces an optimal policy that exhibits behavior which is fully in accordance with the heuristics.

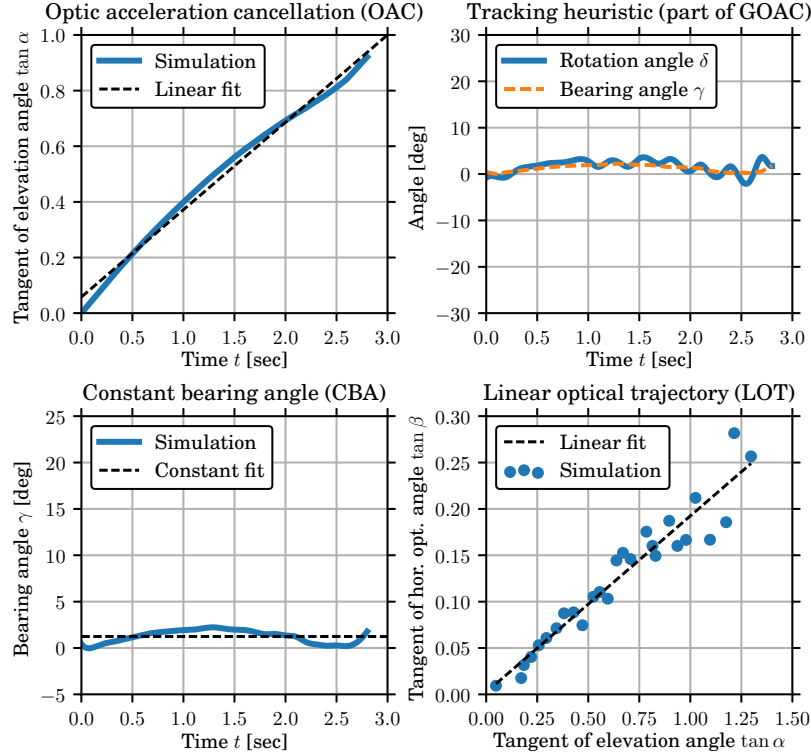

Figure 3: During simulations of successful catches for the continuous tracking scenario encountered by the outfielder (shown in Figure 2), the policies resulting from our optimal control formulation always fulfill the heuristics (OAC, GOAC, CBA, and LOT) from literature with approximately the same precision as in the original human experiments.

## 4.2 Interrupted tracking during long passes—heuristics break but prediction is required

The competing theory to the heuristics claims that a predictive internal model allows humans to intercept the ball [15, 13]. Brancazio [4] points out that "the best outfielders can even turn their backs to the ball, run to the landing point, and then turn and wait for the ball to arrive". Similar behavior is observed in football and american football during long passes. To see whether predictions become necessary, we reproduced situations where the agent cannot catch the ball when acting purely reactively. For example, if the running time to interception point when running backwards (i.e.,

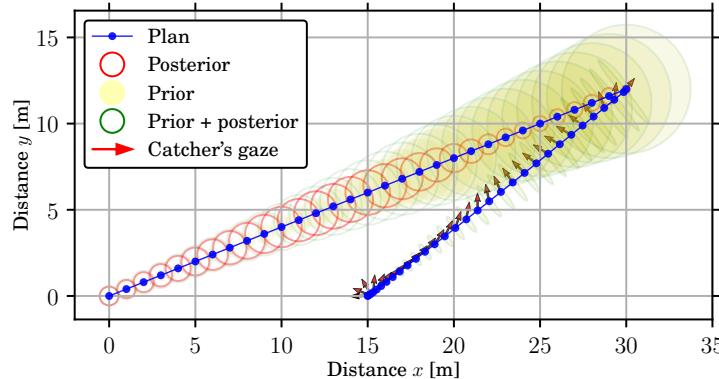

the ratio between the distance to the interception point divided by the maximal backwards running velocity) is substantially higher than the flight time of the ball, no backwards running strategy will be successful. Thus, by varying the initial conditions for the catching agent and the ball, new scenarios can be generated using our optimal control model. The agent's control policy can be tested on reliance on predictions as it is available in form of a computational model, i.e., if the computed policy makes use of the belief states on future time steps, the agent clearly employs an internal model to pre-

Figure 4: An interception plan that leads to successful catch despite violating heuristics. Here, the agent would not be able to reach the interception point in time while running backwards and, thus, has to turn forward to run faster. The resulting optimal control policy relies on beliefs on the future generated by an internal model.

dict the interception point. By choosing appropriate initial conditions for the ball and the agent, we can pursue such scenarios. For example, if the ball flies over the agent's head, he has to turn

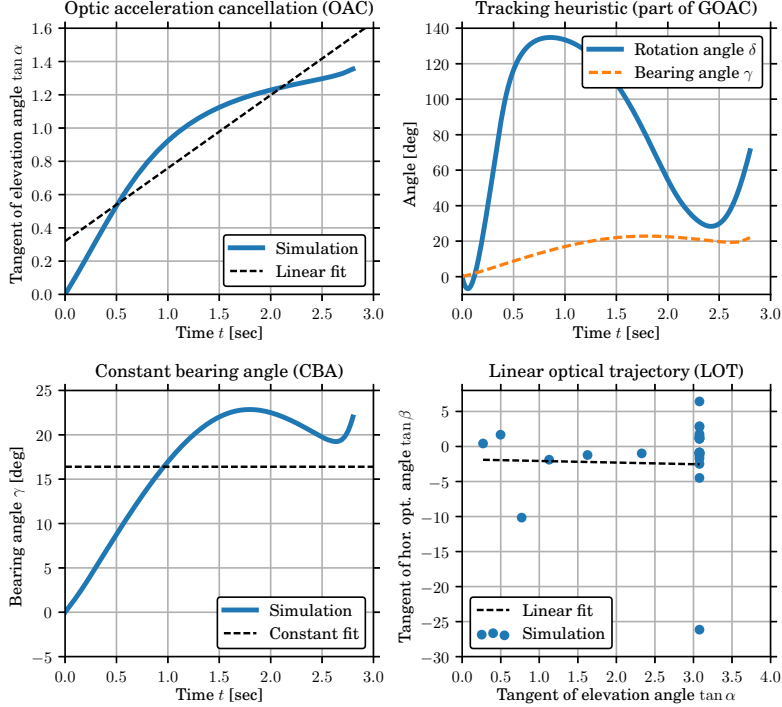

Figure 5: For initial conditions (positions of the ball and the agent) which do not allow the agent to reach the interception point by running backwards or sideways, the optimal policy will include running forward with maximal velocity (as shown in Figure 4). In this case, the agent cannot continuously visually track the ball and, expectedly, the heuristics do not hold.

away from it for a moment in order to gain speed by running forward, instead of running backwards or sideways and looking at the ball all the time. Figure 4 shows such an interception plan where the agent decides to initially speed up and, when sufficiently close, turn around and track the ball while running sideways. Notice that the future belief uncertainty (i.e., the posterior uncertainty $\Sigma$ returned by the extended Kalman filter), represented by red ellipses, grows when the catcher is not looking at the ball and shrinks otherwise. The prior uncertainty (obtained by integrating out future observations), shown in yellow, on the other hand, grows towards the end of the trajectory because future observations are not available at planning time. Similar to [5, 25], we can show for our model predictive control law that the sum of prior and posterior uncertainties (shown as green circles) equals the total system uncertainty obtained by propagating the belief state into the future without incorporating future observations. Figure 5 shows that the heuristics fail to explain this catch—even in the final time steps where the catching agent is tracking the ball to intercept it. OAC deviates from linearity, CBA is not constant, the tracking heuristic wildly deviates from the prediction, and LOT is highly non-linear. GOAC and LOT are affected more dramatically because they directly depend on the catcher's gaze, in contrast to OAC and CBA. Since the heuristics were not meant to describe such situations, they predictably do not hold. Only an internal model can explain the reliance of the optimal policy on the future belief states.

### 4.3 Switching behaviors when uncertainty and reaction time are varied

The previous experiment has pointed us towards policies that switch between predictive subpolicies based on internal models and reactive policies based on current observations. To systematically study what behaviors arise, we use the scenario from Section 4.2 and vary two essential model parameters: system to observation noise ratio $\eta_1 = \log \sigma_b^2/\sigma_o^2$ and reaction time to task duration ratio $\eta_2 = \tau_r/T$, where $T$ is the duration of the ball flight. The system to observation noise ratio effectively determines whether predictions based on the internal model of the dynamics are sufficiently trustworthy for (partially) open-loop behavior or whether reactive control based on the observations of the current state of the system should be preferred. The reaction time to task duration ratio sets the time scale of the problem. For example, an outfielder in baseball may have about $3\,\mathrm{s}$ to catch a ball and his reaction delay of about $200\,\mathrm{ms}$ is negligible, whereas a catcher in baseball often has to act within a fraction of a second, and, thus, the reaction latency becomes crucial.

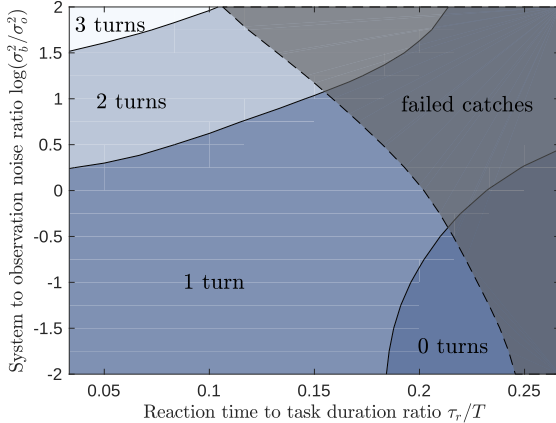

Figure 6: Switches between reactive and feedforward policies are determined by uncertainties and latency.

We run the experiment at different noise levels and time delays and average the results over 10 trials. In all cases, the agent starts at the point $(20, 5)$ looking towards the origin, while the ball flies from the origin towards the point $(30, 15)$ in $3\,\mathrm{s}$. All parameters are kept fixed apart from the reaction time and system noise; in particular, task duration and observation noise are kept fixed. Figure 6 shows how the agent's policy depends on the parameters. Boundaries correspond to contour lines of the function that equals number of times the agent turns towards the ball. We count turns by analyzing trajectories for gaze direction changes and reduction of uncertainty (e.g., in Figure 4 the agent turns once towards the ball). When reaction delays are long and predictions are reliable, the agent turns towards the interception points and runs as fast as he can (purely predictive strategies; lower right corner in Figure 6). When predictions are not sufficiently trustworthy, the agent has to switch multiple times between a reactive policy to gather information and a predictive feedforward strategy to successfully fulfill the task (upper left corner). When reaction time and system noise become sufficiently large, the agent fails to intercept the ball (upper right grayed out area). Thus, seemingly substantially different behaviors can be explained by means of a single model. Note that in this figure a purely reactive strategy (as required for only using the heuristics) is not possible. However, if different initial conditions enabling the purely reactive strategy are used, the upper left corner is dominated by the purely reactive strategy.

## 5    Discussion and conclusion

We have presented a computational model of human interception of a moving target, such as an airborne ball, in form of a continuous state-action partially observable Markov decision problem. Depending on initial conditions, the optimal control solver either generates continuously tracking behavior or dictates the catching agent to turn away from the ball in order to speed up. Interception trajectories in the first case turn out to demonstrate all properties that were previously taken as evidence that humans avoid complex computations by employing simple heuristics. In the second case, we have shown that different regimes of switches between reactive and predictive behavior arise depending on relative uncertainty and latency. When the agent has sufficient time to gather observations (bottom-left in Figure 6), he turns towards the ball as soon as possible and continuously tracks it till the end (e.g., outfielder in baseball acts in this regime). If he is confident in the interception point prediction but the task duration is so short relative to the latency that he does not have sufficient time to gather observations (bottom-right), he will rely entirely on the internal model (e.g., catcher in baseball may act in this regime). If the agent's interception point prediction is rather uncertain (e.g., due to system noise), the agent will gather observations more often regardless of time delays. Conclusions regarding the trade-off between reactive and predictive behaviors may well generalize beyond ball catching to various motor skills. Assuming an agent has an internal model of a task and gets noisy delayed partial observations, he has to tolerate a certain level of uncertainty; if moreover the agent has a limited time to perform the task, he is compelled to act based on prediction instead of observations. As our optimal control policy can explain both reactive heuristics and predictive feedforward strategies, as well as switches between these two kinds of subpolicies, it can be viewed as a unifying explanation for the two seemingly contradictory theories of target interception.

In this paper, we have provided a computational level explanation for a range of observed human behaviors in ball catching. Importantly, while previous interpretations of whether human catching behavior is the result of complex computations or the result of simple heuristics have been inconclusive, here we have demonstrated that what looks like simple rules of thumb from a bag of tricks is actually the optimal solution to a continuous partially observable Markov decision problem. This result therefore fundamentally contributes to our understanding of human rationality.

**Acknowledgements**

This project has received funding from the European Union's Horizon 2020 research and innovation programme under grant agreement No 640554.

## Footnotes

[1] Usain Bolt's world record sprint data http://datagenetics.com/blog/july32013/index.html

[2] World records for backwards running http://www.recordholders.org/en/list/backwards-running.html

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
