[Supplementary Material]

# Model parameters

| Symbol | Description | Value |
|---|---|---|
| $\mathbf{x}_b = [x_b, y_b, z_b]^T$ | Coordinates of the ball | – |
| $\dot{\mathbf{x}}_b = [\dot{x}_b, \dot{y}_b, \dot{z}_b]^T$ | Velocity of the ball | – |
| $\mathbf{x}_c = [x_c, y_c]^T$ | Coordinates of the catcher | – |
| $\dot{\mathbf{x}}_c = [\dot{x}_c, \dot{y}_c]^T$ | Velocity of the catcher | – |
| $\{\phi, \psi\}$ | Spherical angles defining gaze | – |
| $\mathbf{x} = [\mathbf{x}_b, \dot{\mathbf{x}}_b \mathbf{x}_c, \dot{\mathbf{x}}_c, \phi, \psi]^T$ | State of the system | – |
| $\mathbf{z} = [\mathbf{x}_b, \mathbf{x}_c, \phi, \psi]^T$ | Observations | – |
| $\mathbf{u} = [F, \dot{\phi}, \dot{\psi}, \theta]^T$ | Controls | $-2\pi \le \dot{\phi}, \dot{\psi} \le 2\pi,\ |\theta| < \pi$ |
| $\{F_1, F_2\}$ | Force parameters | $\{7.5, 2.5\}$ |
| $\lambda$ | Damping | $5/6$ |
| $\mathbf{Q}$ | System covariance | $\mathrm{diag}\{\sigma_b^2 \mathbf{1}_6, \sigma_c^2 \mathbf{1}_6\}, \sigma_b^2 = 10^{-3}, \sigma_c^2 = 10^{-5}$ |
| $\boldsymbol{\Sigma}_0$ | Initial covariance | $1/4 \cdot \mathrm{diag}\{1/5, 1/5, 0.0, 1/2, 1/2, 0.0, 10^{-2} \mathbf{1}_6\}$ |
| $\{\sigma_{min}, \sigma_{\max}\}$ | Observation noise parameters | $\{10^{-2}, 1.0\}$ |
| $\mathbf{R}$ | Observation covariance | $\mathrm{diag}\{\sigma_o^2 \mathbf{1}_3, \mathbf{0}_4\}$ |
| $\varepsilon_{\mathrm{threshold}}$ | Successful catch distance | $0.5$ |
| $\tau$ | Discretization step | $0.1$ |
| $N$ | Planning horizon length | $N \le 30$ |
| $\{w_0, w_1, w_2, \mathrm{M}\}$ | Cost function weights | $\{10^3, 10^3, 10^2, \mathrm{diag}\{10, 1, 1, 0.1\}\}$ |