[Reviews · NeurIPS 2016]

Reviewer 1

Summary

The authors propose a computational model to explain the behavior of human intercepting moving targets (e.g., balls). This is a classic problem in psychology and there exists multiple rules/heuristics to explain the behavior. Here the model proposed is derived from solving the decision problem at hand given the information available to the agent. It accounts for the existing predictions made by the heuristic but can also handle more complex scenarios.

Qualitative Assessment

The paper is clear and well-written. The choice of decision framework seems appropriate for the task, and the results seem to be of relevance for understanding human strategies. Here are my more detailed comments: - It is said in the paper in a few places that the optimal solution to the control problem is found. This is not the case due to the approximation of doing open-loop planning (Eq 3), even if replanning is done at every step. This may not affect the solution enormously for this particular problem (that depends a lot on the horizon), but it would be nice to emphasize that and discuss what aspects of the optimal policy might be missed with this approximation. - About the fits in Figure 2. It’s good to see that the model can be pushed to reproduce the heuristic predictions, but at what price? There are parameters to the model (e.g., the weighting parameters for the cost, noise params, horizon), how sensitive are the fits/results to changes in these parameters? -I liked Figure 6, really interesting to see the different modes of behavior in this way. -The authors show that the interrupted tracking behavior cannot be modeled with existing heuristics. First, it wasn’t clear to me whether the behavior in this case was matching any human data. Also, while it’s nice to see the model handle these scenarios, it’s not clear that the behavior could not be captured by yet another simple heuristic (naive example: start running forward if some angle related to the ball changes too quickly) without any explicit forward prediction. Why would “turning and running forward for x time” not be accessible to a reactive policy? -Figure 1 is a bit unwieldy. Maybe it’s just me, but I am not sure I understood all the heuristics based on that diagram+ text, maybe it would be better to have a small diagram for each heuristic if feasible?

Confidence in this Review

3-Expert (read the paper in detail, know the area, quite certain of my opinion)


Reviewer 2

Summary

This paper studies how humans catch a ball by modeling the problem at hand as a POMDP. The authors use model predictive control to find ball-catching trajectories and they discuss the different types of solution that occur depending on the parameters of the problem. They find “different regimes of switches between reactive and predictive behavior depending on relative uncertainty and latency”. Interestingly, the reactive policies match the heuristics that are commonly used to describe human ball-catching strategies. This shows that human ball-catching heuristics can be seen as optimal control solutions

Qualitative Assessment

This paper is a fun read with an interesting conclusion. I have very few remarks on the contents and methods. Minor remarks: It would be interested to provide a little more detail on the Model Predictive Control implementation. Did you somehow have to bound the complexity of the problem (e.g. limit the prediction horizon) and/or did you encounter particular issues during the trajectory optimization? How did you pick the parameters w0, w1, w2? The paper is dense and some sections are understandably very concise (e.g. Section 3.2). This might make it harder to reproduce the results. You might consider adding supplementary information with the implementation details.

Confidence in this Review

2-Confident (read it all; understood it all reasonably well)


Reviewer 3

Summary

The authors provide a stochastic optimal control model of catching an airborne ball. By investigating their model in various situations, they show that it provides a unifying explanation that accounts for behaviours generally interpreted as using heuristics in these situations.

Qualitative Assessment

To me, this work is interesting in itself, but its significance is not wide enough to be published in NIPS. The authors should consider taking into account the remarks and then rather submit to a motor control journal. In Section 3.2, many details about the experiment are not explained. Both the ball and the agent are considered as a point, right? This is not stated. What should be the minimal distance between the ball and the agent so that the ball is considered caught? What are the values for all the parameters (F1,F2,lambda, etc.) . They should be given in a table, so that readers can reproduce the experiments. The way Figure 6 is obtained is completely underspecified. How did you get the boundaries of the various areas? How do you count the number of turns in one policy? The caption and the label on axis do not provide exactly the same information. A much more rigorous presentation is required here. Many aspects are modelled using Gaussian functions, but the corresponding assumptions are not discussed. In particular, do you have any convincing argument for assuming beliefs to be Gaussian? In lines 43, 65 and 363, the authors repeat that their study contributes to the question of rationality, but the contribution is not elaborated at all. What do they mean exactly? Lines 149 to 151, the authors should not mix up theoretical presentation and implementation details. The emphasis in line 163 is poorly delimited and useless. The sentence in lines 196 to 199 is quite complex and should be cut.

Confidence in this Review

2-Confident (read it all; understood it all reasonably well)


Reviewer 4

Summary

The paper suggests a model based on partially observable Markov decision processes (POMDPs) for the task of catching a ball. In addition to modeling, this framework shows how common heuristics for ball catching are all an optimal policy when the person can track the ball during the task. Unlike the heuristics, this model can provide the solution for ball catching when the person can not always track the ball visually.

Qualitative Assessment

Unifying all models is definitely very useful for the people working in this field. The paper is easy to follow and the related works are explained very well. The main problem of the paper is that it only contains simulations, not the actual data from a human trying to catch a ball. Are the policies generated from the POMDP model the same as human behavior when the ball is not visually trackable? In sections 3.2, a policy is penalized by the amount of uncertainty explicitly. As mentioned in the paper, an optimal POMDP solver should take care of this naturally. To me, adding this penalty explicitly, looks like a heuristic in a one-step ahead search. How is this different from other heuristics? It seems that other heuristics are based on minimizing uncertainty too (by keeping track of the ball).

Confidence in this Review

2-Confident (read it all; understood it all reasonably well)


Reviewer 5

Summary

This paper attempts to show that previously researched ball catching heuristics can be understood as optimal control policy strategies by modelling catching as a POMDP, and making several assumptions. This paper also explores how varying parameters can affect the optimal strategy in a simulated catching task by varying the task length and the reaction time as well as the noise in the partially observable states. The authors spend a good amount of effort developing a computational model of a human catching a ball with observational uncertainty. This drives their point that heuristics researched in the past may not be sufficient to model scenarios where the reaction time is a significant fraction of task time, or in scenarios where there is a substantial amount of noise.

Qualitative Assessment

This is an interesting paper which explores the catching policies of reaction and planning. While the background is present, it is not presented in a way that fully justifies the experimental simulations. There is some discussion of outfielder vs. catcher catcher scenarios, but those are not fully defined or simulated. A more thorough investigation of these tasks would help to illustrate the simulated model of reaction and planning. While the paper provides generous background and model development the experimental configuration is not fully described, and the results demand additional investigation. This is a cursory introduction to the justification of a model which includes reaction and planning, but additional experimentation would allow for this relationship to be fully elucidated. The statistics are insufficient to show significance and the paper suffers from a lack of proof-reading and editing. Please follow up with these references to further develop the modelling and dynamics as well as the tradeoff between planning and reaction and set ups involving robotic systems: Kim, Seungsu, Ashwini Shukla, and Aude Billard. "Catching objects in flight." IEEE Transactions on Robotics 30.5 (2014): 1049-1065. Zago, Myrka, et al. "Visuo-motor coordination and internal models for object interception." Experimental Brain Research 192.4 (2009): 571-604. Cesqui, Benedetta, et al. "Catching a ball at the right time and place: individual factors matter." PLoS one 7.2 (2012): e31770. Kim, Seungsu, and Aude Billard. "Estimating the non-linear dynamics of free-flying objects." Robotics and Autonomous Systems 60.9 (2012): 1108-1122.

Confidence in this Review

2-Confident (read it all; understood it all reasonably well)